

# Epidemiological profiles and pathogenicity of Vancomycin-resistant *Enterococcus faecium* clinical isolates in Taiwan

Pei-Yun Lin[1], Shang-Yih Chan[2,3,4], Arnold Stern[5], Po-Hsiang Chen[6] and Hung-Chi Yang[7]

[1] Department of Laboratory, Taipei City Hospital, Yang-Ming Branch, Taipei, Taiwan
[2] Department of Internal Medicine, Taipei City Hospital, Yang-Ming Branch, Taipei, Taiwan
[3] Department of Exercise and Health Sciences, University of Taipei, Taipei, Taiwan
[4] Department of Health Care Management, National Taipei University of Nursing and Health Sciences, Taipei, Taiwan
[5] Grossman School of Medicine, New York University, New York, USA
[6] Research Center for Chinese Herbal Medicine, Graduate Institute of Health Industry Technology, College of Human Ecology, Chang Gung University of Science and Technology, Taoyuan, Taiwan
[7] Department of Medical Laboratory Science and Biotechnology, Yuanpei University of Medical Technology, Hsinchu, Taiwan

Corresponding authors
Po-Hsiang Chen,
Bob79620@hotmail.com
Hung-Chi Yang,
hcyang@mail.ypu.edu.tw

## ABSTRACT

The emerging Vancomycin-resistant *Enterococcus faecium* (VRE-fm) is an opportunistic pathogen causing nosocomial infections. The identification of VRE-fm is important for successful prevention and control in healthcare settings. VRE-fm clinical isolates obtained from regional hospitals in northern Taiwan were characterized for antimicrobial susceptibility, virulence genes and biofilm production. Most isolates exhibited multi-drug resistance and carried the virulence genes, *esp* and *hyl*. While all isolates produce biofilms, those isolates that carried *esp* exhibited greater biofilm production. Isolates with different virulence gene carriages were examined for pathogenicity by using a nematode model, *Caenorhabditis elegans*, for determining microbial-host interactions. The survival assay showed that *C. elegans* was susceptible to Linezolid-resistant VRE-fm isolates with *hyl*. Combining the molecular epidemiological profiles regarding pathogenesis in *C. elegans* can serve as a guide for physicians in limiting opportunistic infections caused by VRE-fm.

## INTRODUCTION

*Enterococcus* species are facultative anaerobic Gram-positive bacteria and intestinally commensal in humans. *Enterococci* are well known opportunistic pathogens that cause many infectious diseases, including bacteremia, urinary tract infections (UTI), endocarditis and intra-abdominal or intra-pelvic infections (*Miller et al., 2020*). *Enterococcus faecium* (*E. faecium*) and *Enterococcus faecalis* (*E. faecalis*) are the two most prevalent and clinically related pathogens (*Michaux et al., 2020*). These pathogenic
strains are not the same as those colonized in healthy individuals (*Lee et al., 2019*; *Liese et al., 2019*). Hence, healthcare settings are important in the prevalence and outbreak of *E. faecium* infections.

Antimicrobial therapy is the most effective approach to treat bacterial infections. Increasingly, bacterial pathogens have adapted to a variety of antimicrobials. In particular, multi-drug resistant bacteria have become an urgent issue in healthcare systems (*van Duin & Paterson, 2020*). Compared to *E. faecalis*, *E. faecium* is intrinsically tolerant to a spectrum of antibiotics, including aminoglycosides, β-lactams, cephalosporins and sulphonamides. This property enables *E. faecium* to develop antibiotic resistance in healthcare facilities where antibiotic treatment is common. Since its first appearance in the 1980s, Vancomycin-resistant *Enterococcus faecium* (VRE-fm) has spread world-wide and has received global awareness. Due to its multi-drug resistance, the World Health Organization (WHO) and the Centers for Disease Control and Prevention (CDC) have specified VRE-fm as a priority for developing new drugs (*Tacconelli et al., 2018*).

To establish an infection, *Enterococcus* spp. carry several virulence factors, including aggregation substance (*asa1*), gelatinase (*gelE*), cytolysin (*cylA*), enterococcal surface protein (*esp*) and hyaluronidase (*hyl*) (*Gok et al., 2020*). A product of *asa-1* enhances adherence to cells (*Afonina et al., 2018*). Gelatinase promotes bacterial colonization and spreading by degrading collagen and gelatin (*Pillay, Zishiri & Adeleke, 2018*). Cytolysin increases enterococcal virulence and patient mortality (*Pillay, Zishiri & Adeleke, 2018*). Esp is associated with enhanced virulence, attachment to an abiotic surface, colonization in the urinary tract and biofilm formation (*Jovanovic et al., 2018*). While *esp* is encoded on a pathogenicity island in *E. faecium* and *E. faecalis*, *hyl* is specific to *E. faecium* (*Gok et al., 2020*). *hyl* encodes a putative glycosyl hydrolase, yet, its role in virulence is unclear (*Panesso et al., 2011*).

Biofilms are immobile microbial aggregates that attach to biotic and abiotic surfaces (*Yin et al., 2019*). The bacteria in biofilms are embedded in the extracellular matrix produced by polysaccharides, nucleic acids, lipids and protein (*Powell et al., 2018*). Bacteria establish persistent colonization by producing a biofilm as a shield to resist antibiotics and as an escape from the host immune system and harsh environmental factors. Biofilms are associated with 65% of bacterial infections (*Rather, Gupta & Mandal, 2021b*). In *Enterococcus spp.*, biofilm formation contributes to antimicrobial resistance and virulence (*Cepas et al., 2019*), results in evasion of the host immune system (*Rather, Gupta & Mandal, 2021b*) and facilitates the presence of resistant bacteria in health care facilities (*Luo et al., 2021*). The relationship between biofilm production and enterococci virulence genes remains unclear (*Weng, Ramli & Hamat, 2019*).

Conventional studies of microbial-host interactions have been carried out in mammalian models. Due to regulatory restrictions, many alternative model organisms are available, including nematodes (*Caenorhabditis elegans*), fruit flies (*Drosophila melanogaster*), and zebrafish (*Danio rerio*) (*Kaito et al., 2020*). A free-living organism, *C. elegans*, has been used as an infection model for investigating the virulence of pathogens, including enterococci (*Revtovich et al., 2021*). In this study, *C. elegans* was employed for determining the virulence of VRE-fm clinical isolates with different virulence gene

carriages. The molecular epidemiological profile of VRE-fm clinical isolates and its pathogenesis in *C. elegans* can serve as a guide for physicians in limiting opportunistic infections.

## MATERIALS AND METHODS

### Collection, identification and antimicrobial susceptibility testing of VRE-fm clinical isolates

Sixty clinical isolates of VRE-fm were collected from urine (46), blood (7), pus (4), body fluid (2) or stool (1) from regional hospitals in northern Taiwan. All clinical isolates were cultured overnight on a blood agar plate (Creative Media Plate, New Taipei City, Taiwan) at 37 °C. A single colony of a clinical isolate was inoculated in tryptic soy broth (TSB) (Neogen, Lansing, MI, USA). The overnight culture was adjusted to McFarland 0.5 by using a densitometer. Identification and antimicrobial susceptibility testing were performed by using the BD Phoenix 100 automatic system (BD, Franklin Laker, NJ, USA). The Taipei City Hospital Institutional Review Board granted Ethical approval to carry out the study within its facilities (TCHIRB-10703123-E, Waiver of Informed Consent).

### Multilocus sequence typing (MLST)

MLST is a sequence-based method for establishing the clonal relationship among bacteria (*Bello Gonzalez et al., 2017*). MLST of the VRE-fm clinical isolates was carried out according to previous reports (*Homan et al., 2002*; *Kariyama et al., 2000*). In brief, sequences of seven house-keeping genes specific to *E. faecium*, including *gdh*, *pur*K, *pstS*, *atpA*, *gyd*, *adk*, *ddl*, were amplified by using PCR followed by electrophoresis analysis with 1.2% Agarose gel (100 V, 30 min). PCR products were purified by a commercial Kit (Favorgen, Ping-Tung, Taiwan) and sent for DNA sequencing (Mission Biotech Co., Taipei, Taiwan). The sequencing results were compared with published alleles, and sequence types (STs) were assigned using the MLST database for *E. faecium* (https://pubmlst.org/organisms/enterococcus-faecium/).

### Virulence gene identification

Genomic DNA extraction of the VRE-fm isolates was performed based on a standard protocol (*Kariyama et al., 2000*). In brief, DNA was extracted from an overnight bacterial culture by heating at 95 °C for 5 min followed by centrifugation to remove the debris. All samples were subjected to amplification of the virulence genes (*asa1*, *cylA*, *esp*, *gelE* and *hyl*) by multiplex PCR and DNA electrophoresis as previously described (*Vankerckhoven et al., 2004*). The primer sequences were listed in Table S3. In brief, the PCR mixture was prepared in a total volume of 20 μl containing 2 μl of genomic DNA, 0.5 μl of forward (F) and reverse (R) primer (10 μM), 4 μl of 5X PCR Plus Master Mix II solution (Genemark, Taichung, Taiwan), and 9 μl of distilled water. Multiplex PCR was performed in a GeneAmp PCR System 2700 (Perkin-Elmer, Waltham, MA, USA.). The template was initially denatured at 95 °C for 5 min followed by 30 cycles at 94 °C for 1 min, 60 °C for 1 min, and 72 °C for 1 min. Final extension was set at 72 °C for 10 min. The PCR products

were analyzed by 1.8% agarose gel electrophoresis and visualized by staining with fluorescent dye.

## Biofilm measurements

Biofilm production of the VRE-fm isolates was measured according to a standard protocol with modification (*Hashem, Abdelrahman & Aziz, 2021*). A single colony of clinical isolates was inoculated in TSB supplemented with 1% sucrose and grown at 37 °C. *Staphylococcus epidermidis* strain ATCC35984 was used as a positive control (*Manandhar et al., 2018*). The overnight culture was adjusted to an optical density (OD) at 600 nm with $10^8$ cells/mL in a 96-well culture plate and incubated at 37 °C for 48 h. The culture plate was gently washed thrice with 0.1 ml phosphate-buffered saline (PBS) followed by aspiration and air drying for 30 min (*Weng, Ramli & Hamat, 2019*). 0.1 ml of 0.1% crystal violet was added to the wells for 15 min, followed by washing thrice with 0.1 ml PBS. 0.2 ml of 95% ethanol was added to the wells for 20 min to dissolve the biofilm-associated dye (*Igbinosa & Beshiru, 2019*). The optical density of the samples was determined at 570 nm by using a SpectraMax Max ELISA reader (Molecular Devices, San Jose, CA, USA). A negative control was employed to reduce the background absorbance OD values. Since different clinical isolates had varied growth rates, the resulting bacterial cell number would be different after incubation, leading to inaccurate measurement of the biofilm mass. In order to justify the bias due to cell number, the ability to form a biofilm was expressed by using a biofilm formation parameter: ($OD_{570}$ nm biofilm minus $OD_{570}$ nm control)/$OD_{600}$ nm (cells) (*Lin, Lin & Lan, 2020*). Optical density cut-off (ODc) was defined as three standard deviations above the mean OD of the negative control as described (*Stepanovic et al., 2007*). Each clinical isolate was classified as follows: no-biofilm detected (ND): OD ≤ ODc; weak biofilm producer: ODc < OD ≤ 2 × ODc; moderate biofilm producer: 2 × ODc < OD ≤ 4 × ODc; and strong biofilm producer: OD > 4 × ODc.

## The *C. elegans* culture and survival assay

Wild-type *C. elegans* strain, N2, in their larval stage was cultured with non-pathogenic *Escherichia coli* OP50 as a food source. The *C. elegans* survival assay was performed based on a previous report (*Chen et al., 2020*). Briefly, staged young adult worms were transferred to Tryptic Soy Agar (TSA, Neogen, MI, USA) seeded with a bacterial lawn of Linezolid-resistant VRE-fm clinical isolates and incubated at 20 °C. *E. coli* OP50 were used as a control. Live or dead worms were scored every 24 h. Worms were censored if they crawled off the plate.

## Statistical analysis

Statistics of virulence genes, antibiotic susceptibility and biofilm production were performed with a Chi-square test by using SPSS v. 21 software (SPSS Inc., Chicago, IL, USA). The *C. elegans* lifespan was analyzed by a log-rank test using GraphPad 8 software (GraphPad, San Diego, CA, USA). Statistical significance was indicated by a *P* value (*$P < 0.05$; **$P < 0.01$; ***$P < 0.001$; ns, not significant $P > 0.05$).

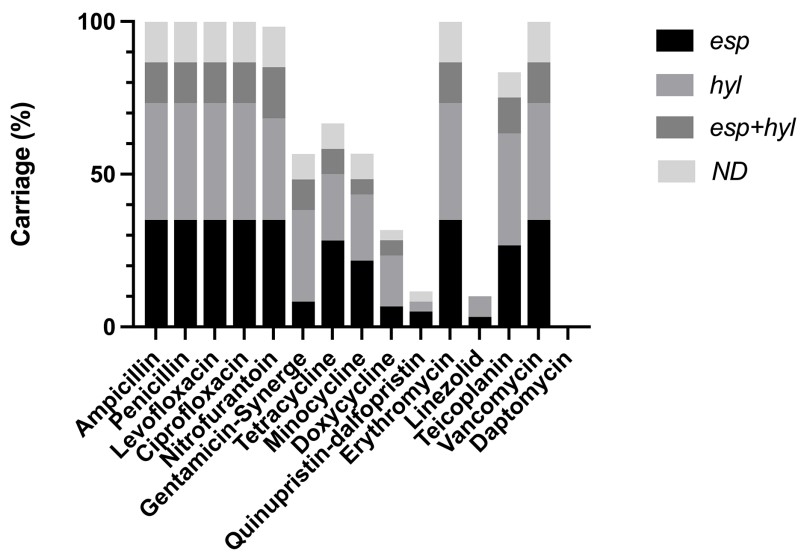

**Figure 1 Distribution of virulence gene carriages in VRE-fm isolates with different antibiotic resistance.** ND: non-detected.

## RESULTS

### Antibiotics susceptibility, molecular typing and virulence gene carriages of VRE-fm clinical isolates

VRE-fm clinical isolates were subjected to analysis of antibiotic susceptibility (Table S1) (Fig. 1). These isolates confer resistance to the most commonly used antibiotics, while being sensitive to a few antibiotics, including Daptomycin, Doxycycline, Linezolid and Quinupristin-dalfopristin. This indicates that a majority of VRE-fm isolates exhibited multi-drug resistance, while a few isolates tolerate Linezolid. Molecular typing based on MLST revealed that three major ST types in clinical isolates, including ST17, ST78 and ST262 (Table 1). The prevalence of ST17 and ST78 is consistent with the findings in a longitudinal study of VRE-fm blood isolates in Taiwan (*Kuo et al., 2018*).

Five virulence genes were tested, including *asa1*, *gelE*, *cylA*, *esp* and *hyl* to determine which enterococcal virulence factor is present in clinical isolates. *asa1*, *gelE* and *cylA* were absent in all isolates, whereas *esp* and *hyl* were detected (Table S2). These isolates can be further classified as $esp^+$ (40.0%, $n = 24$), $hyl^+$ (33.3%, $n = 20$), $esp^+/hyl^+$ (13.3%, $n = 8$) or $esp^-/hyl^-$ (13.3%, $n = 8$) (Table 2). Based on the isolation sites, VRE-fm carrying $esp^+$ and/or $hyl^+$ were isolated mainly from urine (76.6%), followed by blood (11.6%), pus (6.6%), body fluid (3.3%) and stool (1.6%). In urine samples, the percentage of $esp^+$ isolates (39.1%) was higher than that of $hyl^+$ strains (32.6%). The percentage of $esp^+/hyl^+$ isolates (15.2%) was higher than $esp^-/hyl^-$ (13.0%). Nearly all ST17 isolates (95.8%), encoded $esp^+$ or $hyl^+$ (54.2% *vs* 58.3%). In ST78 isolates, $esp^+$ isolates comprised the major group (58.8%), whereas $hyl^+$ isolates were the major group in ST262 isolates (68.4%) (Table 1).

**Table 1 Epidemiological profiles and pathogenicity of Vancomycin-resistant *Enterococcus faecium* clinical isolates in Taiwan analysis of ST types and virulence genes.**

| Virulence gene | ST17 ($n$ = 24) | | ST78 ($n$ = 17) | | ST262 ($n$ = 19) | |
|---|---|---|---|---|---|---|
| | $n$ | % | $n$ | % | $n$ | % |
| *esp* | 13 | 54.2 | 10 | 58.8 | 6 | 31.6 |
| *hyl* | 14 | 58.3 | 4 | 23.5 | 13 | 68.4 |
| *esp+hyl* | 4 | 16.7 | 0 | 0.0 | 4 | 21.1 |
| none | 1 | 4.2 | 3 | 17.6 | 4 | 21.1 |

**Table 2 Virulence gene identification of isolated VRE-fm strains.**

| Isolated site $n$ (%) | | Virulence gene $n$ (%) | | | |
|---|---|---|---|---|---|
| | | *esp* | *hyl* | *esp+hyl* | None of both |
| Urine | 46 (76.6%) | 18 (39.1%) | 15 (32.6%) | 7 (15.2%) | 6 (13.0%) |
| Blood | 7 (11.6%) | 3 (42.8%) | 2 (28.5%) | 1 (14.2%) | 1 (14.2%) |
| Body fluid | 2 (3.3%) | 1 (50.0%) | ND | ND | 1 (50.0%) |
| Stool | 1 (1.6%) | 1 (100.0%) | ND | ND | ND |
| Pus | 4 (6.6%) | 1 (25.0%) | 3 (75.0%) | ND | ND |

**Note:**
   ND: non-detectable.

## The relationship between antibiotic resistance/susceptibility and VRE-fm isolates

Isolates carrying *esp* and/or *hyl* were resistant to antibiotics, including Ampicillin, Penicillin, Levofloxacin, Ciprofloxacin, Erythromycin, and Vancomycin. Some isolates carrying *esp* and/or *hyl* showed either resistance or susceptibility to Doxycyclin, Gentamycin-Synerge, Linezolid, Minocycline, Nitrofurantoin, Quinupristin-dalfopristin, Teicoplanin, and Tetracyclin (Table S2). The *esp* and/or *hyl* were predominant in isolates sensitive to Doxycyclin (68.3%), Quinupristin-dalfopristin (88.3%), Linezolid (90.0%) and Daptomycin (100%) (Table S1). These findings indicate that despite the high degree of the virulence gene carriage, there is a lack of a clear association between antibiotic susceptibility and *esp/hyl* genes in VRE-fm clinical isolates.

## Biofilm production in VRE-fm isolates with an *esp* or *hyl* carriage

Nearly all isolates displayed moderate to strong biofilm production (96.6%, 58/60), while only two isolates showed weak biofilm production. Among the isolates with moderate to strong biofilm production, a majority of isolates carried either *esp* (42%) or *hyl* (42%). While both virulence genes were detected in some isolates (*esp/hyl*, 16%), the absence of both genes were found in a few isolates (*esp⁻/hyl⁻*, 16%). These results indicate that *esp* and/or *hyl* carriage are/is dominant in biofilm producing VRE-fm isolates.

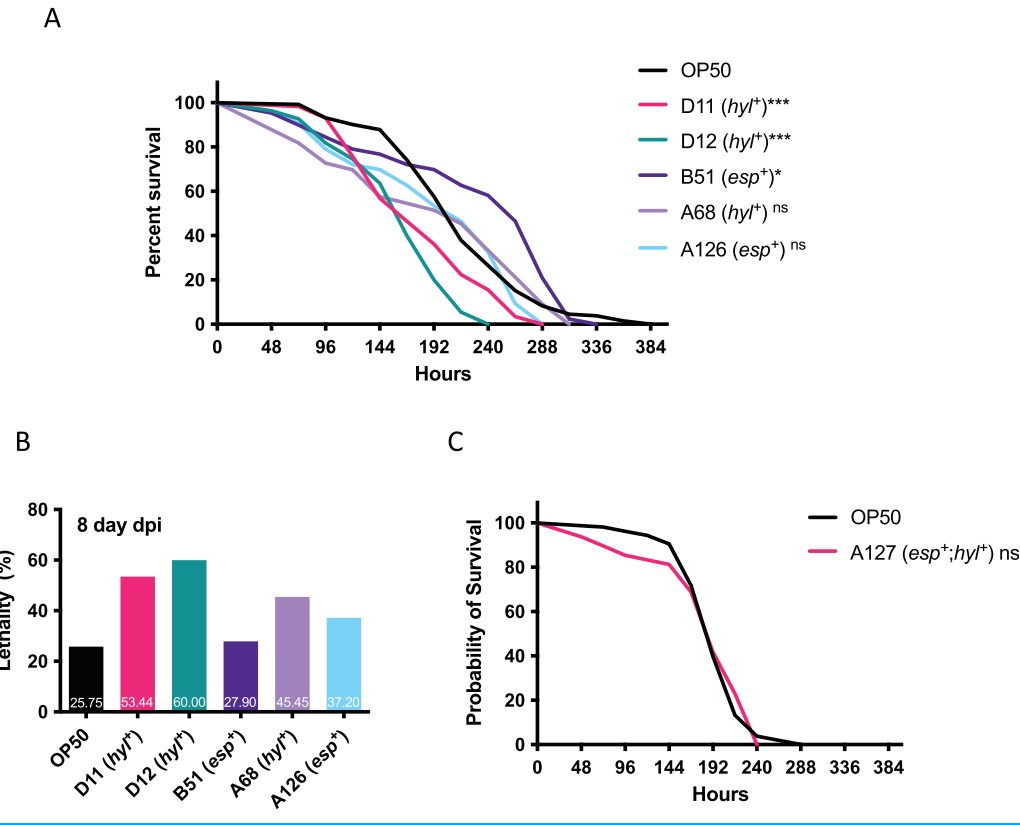

**Figure 2 Survival analysis of Linezolid-resistant/susceptible VRE-fm in the *C. elegans* host.**
(A) Survival analyses were assayed for the *C. elegans* challenged by the indicated Linezolid-resistant
VRE-fm ($n$ = 132 (OP50), 58 (D11), 55 (D12), 43 (B51), 33 (A68), 46 (A126); $^*P < 0.05$, $^{***}P < 0.001$, ns,
$P > 0.05$). (B) Lethality of *C. elegans* at the 8th day after being infected were derived from (A). Percentage
of each condition as indicated. (C) Survival analysis was assayed for *C. elegans* challenged by the
Linezolid-susceptible VRE-fm A127 ($n$ = 48) compared to the *E. coli* OP50 control ($n$ = 53, $P = 0.88$).

## The effect of VRE-fm clinical isolates on *C. elegans* survival

Since multi-drug resistant VRE-fm isolates confer resistance to many clinical antibiotics,
this limits treatment options and requires Linezolid as a last resort. The emerging
Linezolid-resistant VRE-fm is a pressing issue. To this end, Linezolid-resistant VRE-fm
clinical isolates (D11, D12, B51, A68, A126) and a control strain (*E. coli* OP50) were
examined in a *C. elegans* lifespan assay (Fig. 2). Two *hyl*+ strains (D11 and D12) reduced
*C. elegans* lifespan ($P < 0.001$) (Fig. 2A) and survival at the 8th day post infection (Fig. 2B).
While the *C. elegans* lifespan was unaffected by a *hyl*+ strain (A68) ($P = 0.086$), this isolate
increased *C. elegans* lethality at the 8th day post infection (Fig. 2B). Two *esp*+ strains (B51
and A126) were mostly harmless to *C. elegans*.

A *esp*+/*hyl*+ strain with Linezolid susceptibility (A127) was used to determine whether
or not Linezolid resistance in VRE-fm enhances virulence (Fig. 2C). This isolate did not
affect *C. elegans* survival compared to the control ($P = 0.88$), suggesting that resistance to
certain antibiotics, such as Linezolid, as well as the virulence gene, may be required for
VRE-fm pathogenesis.

## DISCUSSION

Bacterial virulence factors support pathogenesis by promoting adhesion, host cell lysis, and antibiotic resistance (*Leitao, 2020*). Pathogenesis of enterococcal infection is achieved partly by the production of virulence factors and resistance to antibiotics. Many enterococcal virulence factors have been identified in *E. faecalis* and *E. faecium*. The distribution of putative virulence markers (PVM) has been proposed for studying the diversity of *E. faecium*. Whether or not these novel putative virulence factors, including Acm (adhesion of collagen from Efm), Scm (second collagen adhesion from Efm), SgrA (serine-glutamine-repeat-containing-protein A) and EcbA (Efm-collagen binding protein A), associate with infection-derived strains is unclear (*Freitas et al., 2018*). The activity of Esp enhances urinary tract adhesion and biofilm production, while the activity of Hyl increases fatality by promoting colonization in the gastrointestinal tract in a murine peritonitis model (*Cho et al., 2018b*). The current study revealed that the high frequency (76.7%) of clinical isolates was found in urine samples, suggesting a preference for VRE-fm colonization in the urinary tract. In a similar virulence gene study with 80 VRE-fm isolates, it is shown that the *esp* carriage (46%) is higher than the *hyl* carriage (20%) (*Arshadi et al., 2018*), while a study with 93 VRE-fm isolates, has shown that the *esp* carriage (60.9%) is higher than the *hyl* carriage (8.7%) (*Say Coskun, 2019*). Consistent with the previous reports, the current study revealed that the *esp* carriage (39.1%) was slightly higher than the *hyl* carriage (32.6%) in urine. The virulence gene carriage in VRE-fm may be associated with UTIs and warrants that VRE-fm be monitored for potential UTIs.

The emerging Linezolid-resistant VRE-fm is a critical issue. Linezolid is a synthetic antibiotic for effective treatment of VRE-fm infections. Although prevalence is low, Linezolid-resistant VRE-fm has been documented in different countries (*Cho et al., 2018a*; *Wardenburg et al., 2019*). Linezolid-resistant VRE-fm may cause higher morbidity and increased medical expenditure than Linezolid-sensitive VRE-fm (*Turner et al., 2021*), suggesting that multi-drug resistance enhances virulence (*Olearo et al., 2021*). Even though there are few Linezolid-resistant VRE-fm clinical isolates (5 in 60 isolates) in the current study, whether or not Linezolid-resistance genes are involved in Linezolid resistance/ sensitivity of VRE-fm remains unclear and warrants further investigation. Forty six Linezolid-sensitive VREFs contain either the *esp* or *hyl* genes, while 6 VREFs contain the *esp*/*hyl* genes with Linezolid resistance (*Say Coskun, 2019*), implying that the relationship between drug resistance and virulence genes remains unclear.

A majority of enterococcal infections are related to biofilm formation, including catheter-related UTI, endocarditis, periodontitis and device-associated infections (*Kunz Coyne et al., 2022*; *Rather et al., 2021a*). Biofilms protect enterococci from the host immune response and antibiotics, thus biofilm-producing enterococci pose a greater risk to disease severity. Biofilm formation confers drug resistance and is associated with *E. faecium* pili (EmpABC) (*Almohamad et al., 2014*; *Fallah et al., 2017*). Despite the fact that several virulence genes are associated with biofilm formation, such as *esp*, it is unclear whether or not *esp* directly contributes to biofilm formation (*Shridhar & Dhanashree, 2019*; *Tendolkar et al., 2004*; *Toledo-Arana et al., 2001*; *Weng, Ramli & Hamat, 2019*).

In *C. elegans*, there are many enterococcal virulence factors reported, including *cyl* (cytolysin), *epaB* (Enterococcal polysaccharide antigen), *fsrA/B/C* (Fsr system), *gelE* (Gelatinase), *lgt* (lipoprotein diacylglyceryl transferase), *paiA* (Transcritional repressor), *phrB* (Deoxyribodipyrimidine photolyase), *recQ* (DNA helicase), *scrB* (Sucrose-6-phosphate hydrolase), and *sprE* (Serine protease) (*Goh et al., 2017*). However, the role of *esp* and *hyl* has not been studied in the *C. elegans* model. A high titer of *E. faecium* can proliferate in the *C. elegans* intestine but it fails to reduce the lifespan of the host (*Revtovich et al., 2021*). In contrast, a low inoculum of *E. faecalis* has caused persistent infection and kills *C. elegans* adults. The killing of the nematode is ascribed to the presence of virulence factors, such as the quorum-sensing system and a cytolysin (*Khan, Jain & Oloketuyi, 2018*). *C. elegans* was mostly affected by the VRE-fm isolates with the *hyl* gene carriage, but not the *esp* gene carriage. This indicates that *hyl* may be associated with virulence during VRE-fm infection in *C. elegans*. A Linezolid-sensitive VRE-fm isolate carrying both the *hyl* and *esp* genes did not reduce the *C. elegans* lifespan, suggesting that both Linezolid resistance and *hyl* are important for VRE-fm pathogenesis. The causal relationship between virulence factors and host survival requires further examination with more VRE-fm clinical isolates as well as mutants of the virulence genes. Since *Enterococci* often cause opportunistic infections, an immunocompromised animal would be a desired host for investigating the exact role of virulence factors in opportunistic infections.

## CONCLUSIONS

VRE-fm clinical isolates obtained from regional hospitals in northern Taiwan were characterized with the intent of understanding the relationship between antimicrobial susceptibility, virulence genes and biofilm production. Most VRE-fm isolates exhibited multi-drug resistance to commonly used antibiotics and carried the virulence genes *esp* and *hyl*. However, there is a lack of association between the specific virulence gene and antibiotic resistance/susceptibility. All VRE-fm isolates were capable of producing biofilms. Isolates carrying *esp* showed greater biofilm production. The host survival assay indicates that *C. elegans* is more sensitive to Linezolid-resistant VRE-fm with *hyl*. This epidemiological information can be beneficial to health care providers in the management of emerging multi-drug resistant microbes. The VRE-fm-*C. elegans* infection model can possibly facilitate an investigation of the molecular mechanisms of novel virulence factors among the emerging multi-drug resistant VRE-fm.

### Funding

This work is mainly supported by a grant from the Department of Health, Taipei City Government (11001-62-031 to Pei-Yun Lin). Support was also received from the Ministry of Science and Technology of Taiwan (MOST-109-2320-B-264-001-MY2 to Hung-Chi Yang). The funders had no role in study design, data collection and analysis, decision to publish, or preparation of the manuscript.

## Grant Disclosures

The following grant information was disclosed by the authors:

Department of Health, Taipei City Government: 11001-62-031 to Pei-Yun Lin.

Ministry of Science and Technology of Taiwan MOST-109-2320-B-264-001-MY2 to Hung-Chi Yang.

## Competing Interests

The authors declare that they have no competing interests.

## Author Contributions

- Pei-Yun Lin conceived and designed the experiments, performed the experiments, analyzed the data, prepared figures and/or tables, and approved the final draft.
- Shang-Yih Chan conceived and designed the experiments, authored or reviewed drafts of the article, and approved the final draft.
- Arnold Stern performed the experiments, authored or reviewed drafts of the article, and approved the final draft.
- Po-Hsiang Chen performed the experiments, analyzed the data, prepared figures and/or tables, and approved the final draft.
- Hung-Chi Yang conceived and designed the experiments, analyzed the data, prepared figures and/or tables, and approved the final draft.

## Ethics

The following information was supplied relating to ethical approvals (*i.e.*, approving body and any reference numbers):

The Taipei City Hospital Institutional Review Board granted Ethical approval to carry out the study within its facilities (Ethical Approval number TCHIRB-10703123-E).

## DNA Deposition

The following information was supplied regarding the deposition of DNA sequences:

The MLST sequences are available at GenBank: BankIt2612181 Seq1 OP244731 to BankIt2612181 Seq21 OP244751.

## Data Availability

The raw data are available in the Supplemental Files.

## Supplemental Information

Supplemental information for this article can be found online at http://dx.doi.org/10.7717/peerj.14859#supplemental-information.

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
