# Peer review of "Epidemiological profiles and pathogenicity of Vancomycin-resistant Enterococcus faecium clinical isolates in Taiwan"

_PeerJ, doi:10.7717/peerj.14859_

## Round 0.1 · original submission · Major Revisions

- This manuscript needs copy editing.
- Several references are before 2018. Replace all of them with newer references. There are several new studies similar to the current study that can be used in the introduction and discussion section.

Reviewer 1 ·

Basic reporting

The basic idea of the article is good and has potential. Nevertheless, the article has some problems.

The virulence genes tested in this work are more commonly found in E. faecalis than in E. faecium. Maybe the authors could look at further E. faecium known virulence/colonization genes PMID: 29149293.

Furthermore, the order that the results are presented. The authors should start presenting the MLST results, then the antimicrobial resistance results and then the virulence genes results.

It could be interesting to test linezolid resistance genes in the few isolates that can tolerate/be resistant to this antibiotic. It would add value to the publication.

Experimental design

There are flaws in the experimental design. In order to reach the conclusions, the authors propose to for Biofilm formation and C. elegans experiments a high number of strains should be used. Furthermore, other types of experiments such as knockouts of the genes in study should be used.

Line 206: Why used linezolid-resistant VREfm? This is not mentioned in the methods section.

Validity of the findings

Line 189: Since all isolates are resistant to vancomycin, being vancomycin-resistant was the selection criteria the authors cannot relate the presence of virulence factors with the presence of resistance to this antibiotic. As there was no possibility to compare the presence of the virulence genes with other vancomycin susceptible isolates.

Lines 202-203: With the results obtained I don't think the authors can reach this conclusion. The type of experiment performed does not allow then reach this conclusion.

Additional comments

Line 46: "Emerging E. faecium may cause nosocomial infections." The previous statement is similar to this one. Could you please elaborate or eliminate this statement?

Lines 94-95: How many isolates were obtained from each sample type? Please include this information in the text.

Reviewer 2 ·

Basic reporting

The manuscript is discussing a very critical problem vancomycin resistant E. faecium giving insight on its virulence determinant, ability to form biofilm and resistance to other antibiotics. However, the manuscript needs English editing using scientific expressions. Numbers in the beginning of sentence must be written in letters.

Experimental design

The experiments are well designed.
Line 121: (or) must be replaced with (and) to adjust the amount of PCR reaction.
In the biofilm measurement, authors did not perform fixation step of the biofilm before CV staining, which is important step. why didn't authors use glacial acetic acid in the resolubilization step instead of ethanol?
How long resolubilization in ethanol did take before measuring the OD??
A statistical analysis to clear the relation between biofilm production and its intensity and the detected virulence determinants especially esp gene is needed.
why did you perform MLST of the isolates??
Are the accession numbers of the sequenced isolates released on the GenBank??

Validity of the findings

Line 172: esp+; hyl+ should be written as esp+/hyl+.
Line 176: The percentage of esp+; hyl+ isolates (15.2%) was similar to esp-; hyl- (13.0%); the mentioned precents could not be considered as similar precents. The same in lines 178-179: "Nearly all ST17 isolates (95.8%), encoded esp+ or hyl+ equally (54.2% vs 58.3%) as 54.2% and 58.3% could not be considered as similar precents.
Line 222: Authors mentioned that "Most enterococcal virulence factors have been identified in E. faecalis, while little is known in E. faecium", this sentence needs reference as E. faecium is well studied.
Lines 227-228: The sentence "The analysis showed that the esp/hyl carriage is common in urine (87%, n=40) and blood samples (86%, n=6)" is as result and should be removed from this section.
Lines 241: "The hyl carrying E. faecium appears more sensitive to vancomycin", Is not the study performed on vancomycin resistant isolates?
Line 242-243: the sentence "46 linezolid sensitive VREFs have either the esp or hyl genes and 6 VREFs have the esp/hyl genes with linezolid resistance (Say Coskun 2019)", why did you cite this reference, isn't it your results?
Line 251-253: You should cite studies that support the association of biofilm formation with presence of esp gene and the other studies that does not.
Regarding The effect of VRE-fm clinical isolates on C. elegans survival, the results are not clearly displayed. for example, D11 and D12 reduced C. elegans lifespan (P<0.001) and the survival at the
8th day post infection, and in the figure 2A the strain D12 takes more than 8 days while D11 takes 12 days approximately. Specify the 8th day at which you make the measurements on the figure.
results of MLST were not discussed or interpretated.
How could be linezolid resistance important for VEE. faecium pathogenesis??

---

## Round 0.2 · accepted · Accept

Dear Dr. Yang,
Thank you for your submission to PeerJ.

I am writing to inform you that your manuscript - Epidemiological profiles and pathogenicity of Vancomycin-resistant Enterococcus faecium clinical isolates in Taiwan - has been Accepted for publication. Congratulations!